

# Efficient higher-order matrix product operators for time evolution

**Maarten Van Damme[1], Jutho Haegeman[1],**
**Ian McCulloch[2,3,4] and Laurens Vanderstraeten[1,5⋆]**

**1** Department of Physics and Astronomy, University of Ghent, Belgium
**2** School of Mathematics and Physics, The University of Queensland, Australia
**3** Frontier Center for Theory and Computation, National Tsing Hua University,
Hsinchu 30013, Taiwan
**4** Department of Physics, National Tsing Hua University, Hsinchu 30013, Taiwan
**5** Center for Nonlinear Phenomena and Complex Systems,
Université Libre de Bruxelles, Belgium

⋆ laurens.vanderstraeten@ulb.be

## Abstract

We introduce a systematic construction of higher-order matrix product operator (MPO) approximations of the time evolution operator for generic (short and long range) one-dimensional Hamiltonians. We demonstrate the utility of our construction, by showing an order of magnitude speedup in simulation cost compared to conventional first-order MPO time evolution schemes.



## 1  Introduction

Some years following the discovery of the density matrix renormalization group (DMRG) [1] algorithm, it was reformulated as a variational method in the language of matrix product states (MPS). This proved to be a fruitful endeavor, as it not only explained the astounding accuracy of DMRG in approximating ground state properties of strongly interacting one-dimensional quantum systems, but it also opened the door to a zoo of algorithms that greatly extend the range of applicability beyond mere ground state properties [2].

In particular, it was realized that MPS can also be used to simulate the time evolution of an interacting system. Although the entanglement in a state generically increases under unitary time evolution and the MPS bond dimension would have to grow exponentially, in practice MPS simulations can reach surprisingly long times with high accuracy. Initial algorithms were limited to short-range interacting systems by using the Trotter-Suzuki decomposition of the time-evolution operator [3–5]. This restriction has by now been lifted using more involved algorithms [6–8], allowing one to target even quasi two-dimensional and long-range interacting systems. Still, these methods all rely on evolving states by taking small time steps, to the effect that some non-equilibrium properties remain difficult to calculate up to the desired precision without investing a tremendous amount of CPU or GPU hours. Recently, a new approach [9] based on cluster expansions was introduced to find tensor network approximations of the time evolution operator that are accurate for much larger time steps, but again this approach is limited to short-range interactions.

In this work, we introduce an approach based on matrix product operators (MPO) [10] that allows us to approximate the full time-evolution operator up to arbitrary order, even for long-range interactions. Our construction can be seen as a higher-order generalization of the $W_I/W_{II}$ operators of Ref. [7] or as an extension of the cluster-expansion approach of Ref. [9] to generic Hamiltonians; the form of the MPO reduces to the one of Ref. [7] when considering the first-order case. We demonstrate the utility of such a higher-order scheme in practice, as it is shown to drastically outperform state-of-the-art algorithms for simulating time evolution with MPS.

The resulting algorithm is both simple to implement and highly flexible, applicable to both finite and infinite systems with arbitrary unit cells and non-abelian symmetries, as long as the Hamiltonian can be represented as an MPO [11]. We provide an example implementation and include analytical MPO expressions that can be implemented and combined with pre-existing tensor network toolboxes.

## 2 Matrix product states and matrix product operators

In this first section we recapitulate all the essentials on MPS and MPO representations, in order to fix notation and set the stage for the next sections.

### 2.1 General notation

A matrix product state (MPS) is represented as

$$|\Psi\rangle_{\text{finite}} = \boxed{M_1}\!-\!\boxed{M_2}\!-\!\boxed{M_3}\!-\!\boxed{M_4}\!-\!\boxed{M_5}\,, \tag{1}$$

where the variational parameters are contained within the local complex-valued three-leg tensors $M^i$. The dimensions of the virtual bonds of the MPS tensors are called the bond dimension. Similarly as for states, a matrix product operator (MPO) can be constructed as the contraction of local four-leg tensors

$$O_{\text{finite}} = \boxed{O_1}\!-\!\boxed{O_2}\!-\!\boxed{O_3}\!-\!\boxed{O_4}\!-\!\boxed{O_5}\,. \tag{2}$$

This representation of a quantum state is size-extensive, in the sense that the state is built up from local objects. The construction can therefore be extended to an infinite system, where the state is built up as an infinite repetition of an $n$-site unit cell of tensors $M^i$:

$$|\Psi\rangle_{\text{infinite}} = \cdots -\boxed{M_1}\!-\!\boxed{\cdots}\!-\!\boxed{M_n}\!-\!\boxed{M_1}\!-\!\boxed{\cdots}\!-\!\boxed{M_n}\!- \cdots\,. \tag{3}$$

The norm of such an infinite-system state is given by

$$\cdots \begin{array}{c}\boxed{M_1}\!-\!\boxed{\cdots}\!-\!\boxed{M_n}\!-\!\boxed{M_1}\!-\!\boxed{\cdots}\!-\!\boxed{M_n}\\[2pt]\boxed{\bar{M}_1}\!-\!\boxed{\cdots}\!-\!\boxed{\bar{M}_n}\!-\!\boxed{\bar{M}_1}\!-\!\boxed{\cdots}\!-\!\boxed{\bar{M}_n}\end{array} \cdots\,, \tag{4}$$

and is well-defined if the unit-cell transfer matrix,

$$\begin{array}{c}\boxed{M_1}\!-\!\boxed{\cdots}\!-\!\boxed{M_n}\\[2pt]\boxed{\bar{M}_1}\!-\!\boxed{\cdots}\!-\!\boxed{\bar{M}_n}\end{array} \tag{5}$$

has a unique leading eigenvalue – this is called an injective MPS. In that case the leading eigenvalue is necessarily real-positive, and we can naturally normalize by rescaling the MPS tensors such that the leading eigenvalue of the transfer matrix is set to one.[1]

An MPO can be similarly considered directly in the thermodynamic limit, and the expectation value of this MPO with respect to an MPS is characterized by the leading eigenvalue of the triple-layer transfer matrix

$$\lambda = \rho_{\max}\left(\begin{array}{c}\boxed{M_1}\!-\!\boxed{\cdots}\!-\!\boxed{M_n}\\[2pt]\boxed{O_1}\!-\!\boxed{\cdots}\!-\!\boxed{O_n}\\[2pt]\boxed{\bar{M}_1}\!-\!\boxed{\cdots}\!-\!\boxed{\bar{M}_n}\end{array}\right)\,, \tag{6}$$

such that we can evaluate

$$\lambda = \lim_{N\to\infty}\frac{1}{N}\log(\langle\Psi|O|\Psi\rangle) \tag{7}$$

(where $N$ denotes the diverging system size). If this triple-layer transfer matrix is diagonalizable and has a unique leading eigenvalue, the MPO is called a zero-degree MPO.[2]

---

[1]For more details on uniform MPS, we refer the reader to Ref. [12].

[2]Here we take a simple definition of an $n$'th degree MPO, which is related to the scale of the norm of the MPO

Table 1: Different methods for applying an MPO to an MPS.

| algorithm | scaling | finite | infinite | iterative |
|---|---|---|---|---|
| naive [2] | $O(D^3\chi^3)$ | ✓ | ✓ | |
| zip-up [6,17] | $O(D\chi^3)$ | ✓ | | |
| density matrix algorithm [18] | $O(D^2\chi^3)$ | ✓ | ✓ | |
| (i)DMRG [13,14] | $O(D\chi^3)$ | ✓ | ✓ | ✓ |
| non-linear optimization | $O(D\chi^3)$ | ✓ | ✓ | ✓ |
| variational uniform MPS [16] | $O(D\chi^3)$ | | ✓ | ✓ |

## 2.2 Applying an MPO to an MPS

One of the most basic steps in MPS-based algorithms is the application of an MPO to an MPS. The bond dimension of the resulting MPS is the product of the original MPS and MPO bond dimensions, which becomes intractable after doing a few consecutive MPO applications. Therefore, we want to approximate the result again as an MPS with a smaller bond dimension:

$$\tag{8}$$

A natural way to find the tensors $M_i'$ is to naively apply the MPO of bond dimension $D$ to the MPS of bond dimension $\chi$, yielding an MPS with bond dimension $\chi' = \chi D$. In a second step we can then truncate this bond dimension down using the Schmidt decomposition, giving an algorithm scaling as $O(D^3\chi^3)$.

There are more performant schemes available, for example by directly minimizing the 2-norm difference between the left and right hand side of Eq. 8. For finite systems this can be done by a sweeping-like DMRG scheme [13,14] or with a global non-linear optimization scheme [15] – the latter can be extended to infinite systems by using variational schemes over uniform MPS [12,16]. Alternatively, for finite systems there is the zip-up method [6,17] that performs singular-value decompositions without first bringing the state into canonical form. This softens the computational cost considerably, and only leads to small errors. Finally, there is a method based on consecutive truncations of the reduced density matrix [18], also yielding a smaller computational costs. In Table 1 we summarize these different methods with their scope and computational costs. The benchmarks in Sec. 8 were always performed using variational schemes.

## 2.3 MPO representation of extensive Hamiltonians

A generic spin-chain Hamiltonian $H$ can be represented as an MPO, with the local MPO tensor having the following substructure [2,19,20]:

$$H \sim \begin{pmatrix} \mathbb{I} & C & D \\ & A & B \\ & & \mathbb{I} \end{pmatrix}. \tag{9}$$

---

in the thermodynamic limit. Since the choice of norm for an operator is not fixed naturally as it is for states, we do not go into detail here on this definition in terms of operator norms – we refer to Ref. [11] for more details. Here, it suffices to refer to the scaling of the expectation value of the MPO with respect to an injective MPS.

The blocks $A$, $B$, $C$ and $D$ are all four-leg tensors and $\mathbb{I}$ is the identity operator acting on the local Hilbert space:

$$\mathbb{I} = \cdots\rangle\cdots \, , \qquad A = {}^{\chi}\Box^{\chi} \, , \qquad C = \cdots\Box^{\chi} \, , \qquad \qquad (10)$$
$$B = {}^{\chi}\Box\cdots \, , \qquad D = \cdots\Box\cdots \, .$$

The dimensions of the first and last virtual levels is always one (denoted by the dashed line above), but the dimension of the middle level can be larger; this dimension is henceforth called the MPO's bond dimension $\chi$. We always require that the spectral radius[3] of the middle block $A$ is smaller than one.

This operator is a first-degree MPO [11], in the sense that the expectation value with respect to an injective MPS scales linearly with system size – as it should for a local Hamiltonian. This is reflected in the structure of the triple-layer transfer matrix [Eq. 6], which has a unique dominant eigenvalue with value 1 (provided the MPS is properly normalized), to which is associated a two-dimensional generalized eigenspace, or thus, a two-dimensional Jordan block. Upon taking the $N$th power, this gives rise to terms scaling as $1^N$, thus constant, as well as terms scaling as $N 1^N$, or thus linearly in $N$. The prefactor of this last term corresponds exactly to the bulk energy density.

A particularly insightful way of representing a first-degree MPO is by a finite-state machine [21]:

$$ \qquad \qquad (11)$$

which makes the meaning of the different blocks immediately clear: When going from left to right through the MPO, the virtual level '1' denotes that the Hamiltonian has not yet acted, the virtual level '2' denotes that the Hamiltonian is acting non-trivially and the virtual level '3' denotes that the Hamiltonian has acted completely. Transitions between the levels are performed in the MPO by the non-trivial blocks. Contracting the MPO from left to right, one can never go down a level.

Written out in full, the Hamiltonian is given by

$$H = \sum_i \left( D_i + C_i B_{i+1} + C_i A_{i+1} B_{i+2} + C_i A_{i+1} A_{i+2} B_{i+3} + \dots \right) . \qquad (12)$$

This shows that any Hamiltonian with exponentially decaying interactions can be efficiently represented by an MPO of this form. Moreover, other decay profiles can often be very well approximated by this type of MPO [10, 11].

## 2.4 Examples

It is instructive to give a few examples of Hamiltonians written in this form, partly because we will use these examples as benchmark cases in Sec. 8. The nearest-neighbour transverse-field Ising model is defined by the Hamiltonian

$$H_{\text{ising,nn}} = -\sum_i Z_i Z_{i+1} + h \sum_i X_i \sim \begin{pmatrix} \mathbb{I} & -Z & hX \\ & 0 & Z \\ & & \mathbb{I} \end{pmatrix} . \qquad (13)$$

---

[3]Here 'spectral radius' is again interpreted in terms of the triple-layer transfer matrix with respect to an injective MPS, now restricted to the diagonal $A$ block; for more details on the conditions on the MPO, we again refer to Ref. [11].

In this case, the diagonal $A$ block is zero and the dimension of the middle level is $\chi = 1$. This Hamiltonian can be extended with long-range exponentially-decaying interactions by including an entry on the diagonal

$$H_{\text{ising,lr}} = -\sum_{i<j} \lambda^{j-i-1} Z_i Z_j + h\sum_i X_i \sim \begin{pmatrix} \mathbb{I} & -Z & hX \\ & \lambda\mathbb{I} & Z \\ & & \mathbb{I} \end{pmatrix}, \qquad \lambda < 1. \tag{14}$$

Another paradigmatic example is the Heisenberg spin-1/2 chain, represented as

$$H_{\text{heisenberg,nn}} = \sum_i S_i^\alpha S_j^\alpha \sim \begin{pmatrix} \mathbb{I} & S^\alpha & 0 \\ 0 & & S^\alpha \\ & & \mathbb{I} \end{pmatrix}. \tag{15}$$

Here, the spin operators are $S^\alpha = (S^x, S^y, S^z)$, such that the blocks have dimension $\chi = 3$.[4] A next-nearest-neighbour $J_1$-$J_2$ spin-1/2 chain is given by

$$H_{\text{heisenberg,nnn}} = J_1 \sum_i S_i^\alpha S_{i+1}^\alpha + J_2 \sum_i S_i^\alpha S_{i+2}^\alpha \sim \begin{pmatrix} \mathbb{I} & S^\alpha & 0 & 0 \\ 0 & \mathbb{I} & J_1 S^\alpha \\ 0 & 0 & J_2 S^\alpha \\ & & & \mathbb{I} \end{pmatrix}, \tag{17}$$

where the tensor $\mathbb{I}$ in the $A$ block again represents the direct product of two unit matrices,

$$\mathbb{I} = \;\; \text{---}\rangle\text{---} \;. \tag{18}$$

Finally, we give an example of a two-dimensional system, which we have wrapped onto a cylinder such that the model can be reformulated as a one-dimensional system. The transverse-field Ising model on a square lattice formulated on a cylinder of circumference $L_y$ with spiral boundary conditions is given by

$$H_{\text{ising,cylinder}} = -\sum_i Z_i Z_{i+1} - \sum_i Z_i Z_{i+L_y} + h\sum_i X_i \sim \begin{pmatrix} \mathbb{I} & -Z & 0 & 0 & \ldots & 0 & hX \\ 0 & \mathbb{I} & 0 & \ldots & 0 & Z \\ 0 & 0 & \mathbb{I} & \ldots & 0 & 0 \\ \vdots & & & \ddots & & \vdots \\ 0 & & & \ldots & \mathbb{I} & 0 \\ 0 & & & \ldots & 0 & Z \\ & & & & & \mathbb{I} \end{pmatrix}. \tag{19}$$

---

[4]Without encoding SU(2) symmetry explicitly in the MPO, it can be simply rewritten in the form

$$\begin{pmatrix} \mathbb{I} & S^x & S^y & S^z & 0 \\ 0 & 0 & 0 & S^x \\ 0 & 0 & 0 & S^y \\ 0 & 0 & 0 & S^z \\ & & & & \mathbb{I} \end{pmatrix}.$$

When encoding SU(2) symmetry, however, we cannot split up the MPO into spin components (which break SU(2) invariance) and we have to keep the above form with the $S^\alpha$ tensor defined as

$$\boxed{\phantom{xxx}} = \boxed{S}\!-\!\overset{\alpha}{\phantom{x}}\!-\!\boxed{S}, \qquad S^\alpha = \boxed{S}\!-\!\overset{\alpha}{\phantom{x}}. \tag{16}$$

Here, the leg denoted by $\alpha$ transforms under the spin-1 representation of SU(2).

## 2.5 Powers of MPOs

This MPO representation of Hamiltonians is convenient for expressing powers of the Hamiltonian, and evaluating e.g. the variance or higher-order cumulants of the Hamiltonian with respect to a given MPS.

We start by rewriting the Hamiltonian in table form:

$$
H \sim
\begin{array}{c|c|c|c}
 & (1) & (2) & (3) \\
\hline
(1) & \mathbb{I} & C & D \\
\hline
(2) & & A & B \\
\hline
(3) & & & \mathbb{I}
\end{array}
\tag{20}
$$

We can now represent $H^2$, the product of this Hamiltonian with itself, as a sparse MPO of the form

| | (1,1) | (1,2) | (1,3) | (2,1) | (2,2) | (2,3) | (3,1) | (3,2) | (3,3) |
|---|---|---|---|---|---|---|---|---|---|
| (1,1) | $\mathbb{I}$ | $C$ | $D$ | $C$ | $CC$ | $CD$ | $D$ | $DC$ | $DD$ |
| (1,2) | | $A$ | $B$ | | $CA$ | $CB$ | | $DA$ | $DB$ |
| (1,3) | | | $\mathbb{I}$ | | | $C$ | | | $D$ |
| (2,1) | | | | $A$ | $AC$ | $AD$ | $B$ | $BC$ | $BD$ |
| (2,2) | | | | | $AA$ | $AB$ | | $BA$ | $BB$ |
| (2,3) | | | | | | $A$ | | | $B$ |
| (3,1) | | | | | | | $\mathbb{I}$ | $C$ | $D$ |
| (3,2) | | | | | | | | $A$ | $B$ |
| (3,3) | | | | | | | | | $\mathbb{I}$ |

$$\tag{21}$$

Here we have used a particular notation for combining the blocks: we take the operator product (composition) on the physical legs and a direct product on the virtual legs. For example:

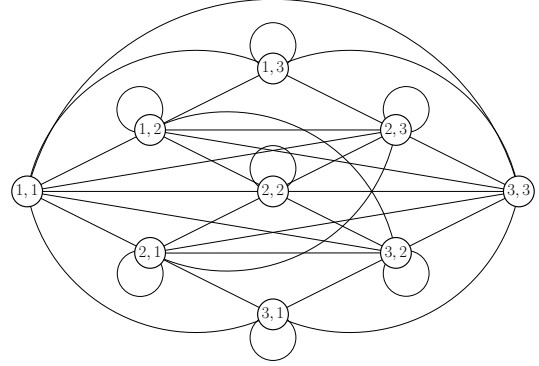

$$\tag{22}$$

Upon computing the triple-layer transfer matrix associated to taking the expectation value of $H^2$ with respect to an injective MPS, the diagonal blocks $\mathbb{I}$ in the above form will give rise to an eigenvalue 1, which will have an (algebraic) multiplicity of 4. For reasons to be explained in Section 4, this will decompose into a one-dimensional eigenspace that does not couple to the boundary conditions, and a three-dimensional generalised eigenspace, or thus a three-dimensional Jordan block, giving rise to terms scaling as a second order polynomial of $N$ upon taking the $N$th power. It therefore represents a second-degree MPO and its expectation value can be evaluated using the methods of Refs. [19, 22]. Again, we can understand this MPO as a finite-state machine

where we have omitted the operators denoting the different transitions in the graph (they can be read off from the table). The structure of this MPO is best understood by decomposing it into two parts, i.e. the disconnected terms and the connected terms. The former are the terms that are the direct product of single actions of the Hamiltonians that do not overlap, and in the diagram they are obtained by passing through levels (1,3) or (3,1). Indeed, the meaning of these levels is that one of the Hamiltonians has already acted, whereas the second one has not. The connected terms are the ones where the two Hamiltonian operators overlap. For example, jumping from (1,1) or (1,2) immediately to (2,3) means that the two Hamiltonian operators overlap on one site, and similarly for the jump from (1,1) or (2,1) to (3,2). All the other connected terms pass through level (2,2), which denotes that both Hamiltonian operators are acting simultaneously, and therefore this level has a bond dimension $\chi^2$.

## 3    From powers of the Hamiltonian to extensive MPOs

Let us now investigate how to approximate the exponential of the Hamiltonian in terms of MPOs. We take a generic spin-chain Hamiltonian $H = \sum_i h_i$, with $h_i$ the (quasi) local hamiltonian operator acting on sites $i, i+1, \ldots$, which can be represented as an MPO of the form in Eq. (20). We wish to approximate

$$\mathrm{e}^{\tau H} = \mathbb{I} + \tau H + \frac{\tau^2}{2}H^2 + \frac{\tau^3}{6}H^3 + \ldots, \tag{23}$$

where we assume that $\tau$ is a small parameter. Naively, one could try to use the above representation of $H^n$ to approximate the exponential. Adding different powers of $H$ is, however, an ill-defined operation in the thermodynamic limit because the norms of these different terms scale with different powers of system size. Therefore, applying a sum of different powers of $H$ to a given state $|\Psi\rangle$, would yield a state

$$\mathrm{e}^{\tau H}|\Psi\rangle \approx \sum_{i=0}^{N}|\Psi_i\rangle, \qquad \langle\Psi_i|\Psi_i\rangle \propto N^i, \tag{24}$$

which cannot be normalized in the thermodynamic limit.[5]

Instead, an appropriate MPO representation of $\mathrm{e}^{\tau H}$ requires a size-extensive approach. Therefore, we introduce a transformation that maps a given power of $H$ to a size-extensive operator, yielding an $n$'th order approximation for $\mathrm{e}^{\tau H}$. We start at first order. Given the finite-state machine representation of $H$, the transformation can be visualized as

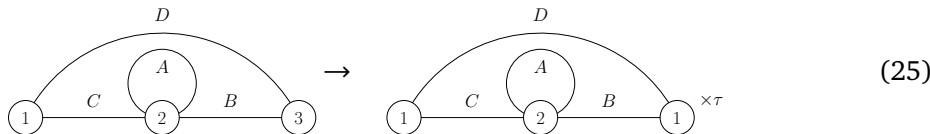

$$\tag{25}$$

I.e., instead of falling onto the level '3' in the MPO for the Hamiltonian, we go back to level '1' and we omit level '3' from the MPO. In addition, we multiply with the appropriate factor $\tau$. In table form, this gives rise to

|     | (1)                 | (2) |
| --- | ------------------- | --- |
| (1) | $\mathbb{I} + \tau D$ | $C$ |
| (2) | $\tau B$            | $A$ |

$$\tag{26}$$

---

[5]This problem suggests that MPS methods that rely on taking powers of the Hamiltonian do not scale well for large system sizes and cannot be formulated directly in the thermodynamic limit.

which serves as a first-order approximation of the time evolution operator $e^{\tau H}$, as introduced in Ref. [7]. In the absence of any Jordan blocks, this operator is size-extensive: upon applying this MPO to a normalizable state, it returns a normalizable state. It is also size-extensive in another sense: it contains all disconnected higher-order terms in the expansion (with correct prefactor), i.e. higher order terms in which different actions of the Hamiltonian do not overlap. Indeed, if we write out the MPO from Eq. (26) in orders of $\tau$ we obtain

$$\mathbb{I} + \tau \sum_i h_i + \tau^2 \sum_{i<j,\text{disc}} h_i h_j + \tau^3 \sum_{i<j<k,\text{disc}} h_i h_j h_k + \dots , \tag{27}$$

where the second and third sum runs over all terms for which the $h_i$ do not overlap.

This transformation can be extended to second order, where we have to include the terms where two actions of the Hamiltonian overlap. These are contained within the MPO representation of $H^2$ [Eq. (21)], so this is the starting point. The level (1,3) encodes the situation where one action of the Hamiltonian has been applied, while the other Hamiltonian can be recognized in the subblock

|       | (1,3)        | (2,3) | (3,3)        |
|-------|--------------|-------|--------------|
| (1,3) | $\mathbb{I}$ | $C$   | $D$          |
| (2,3) |              | $A$   | $B$          |
| (3,3) |              |       | $\mathbb{I}$ |

$$\tag{28}$$

This level (1,3) therefore encodes a disconnected term in $H^2$ and should be immediately mapped back to the starting state (1,1). The (3,1) level is completely equivalent to the (1,3) level, and should also be mapped back to the starting state (1,1). In practice this can be done by taking the columns (1,3) and (3,1) in $H^2$, multiplying by $\frac{\tau}{2}$, and adding them to the first column. Afterwards both columns are removed, and we end up with the MPO:

|       | (1,1)                  | (1,2) | (2,1) | (2,2) | (2,3) | (3,2) | (3,3)        |
|-------|------------------------|-------|-------|-------|-------|-------|--------------|
| (1,1) | $\mathbb{I} + \tau D$  | $C$   | $C$   | $CC$  | $CD$  | $DC$  | $DD$         |
| (1,2) | $\frac{\tau}{2}B$      | $A$   |       | $CA$  | $CB$  | $DA$  | $DB$         |
| (2,1) | $\frac{\tau}{2}B$      |       | $A$   | $AC$  | $AD$  | $BC$  | $BD$         |
| (2,2) |                        |       |       | $AA$  | $AB$  | $BA$  | $BB$         |
| (2,3) |                        |       |       |       | $A$   |       | $B$          |
| (3,2) |                        |       |       |       |       | $A$   | $B$          |
| (3,3) |                        |       |       |       |       |       | $\mathbb{I}$ |

$$\tag{29}$$

In terms of the finite-state machine, one can think of this operation as follows

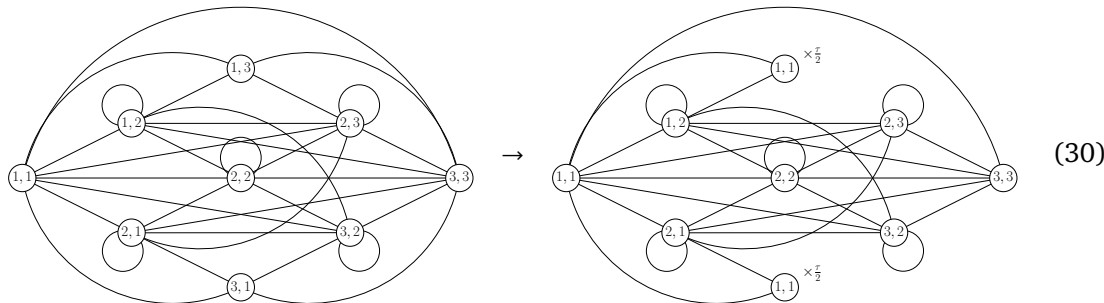

$$\tag{30}$$

The (3,3) level represents the state where both Hamiltonians were applied. Because we have already filtered out the disconnected contributions in the previous step, this state now only contains the connected second-order contributions! Similar to the (1,3) case, we can take the

(3,3) column, this time multiply by $\frac{\tau^2}{2}$, and add it to the first column. Then remove the (3,3) row and column:

| | | (1,1) | (1,2) | (2,1) | (2,2) | (2,3) | (3,2) |
|---|---|---|---|---|---|---|---|
| | (1,1) | $\mathbb{I} + \tau D + \frac{\tau^2}{2}DD$ | $C$ | $C$ | $CC$ | $CD$ | $DC$ |
| | (1,2) | $\frac{\tau}{2}B + \frac{\tau^2}{2}DB$ | $A$ | | $CA$ | $CB$ | $DA$ |
| | (2,1) | $\frac{\tau}{2}B + \frac{\tau^2}{2}BD$ | | $A$ | $AC$ | $AD$ | $BC$ |
| | (2,2) | $\frac{\tau^2}{2}BB$ | | | $AA$ | $AB$ | $BA$ |
| | (2,3) | $\frac{\tau^2}{2}B$ | | | | $A$ | |
| | (3,2) | $\frac{\tau^2}{2}B$ | | | | | $A$ |

(31)

or in terms of a finite-state machine, we take the transformation

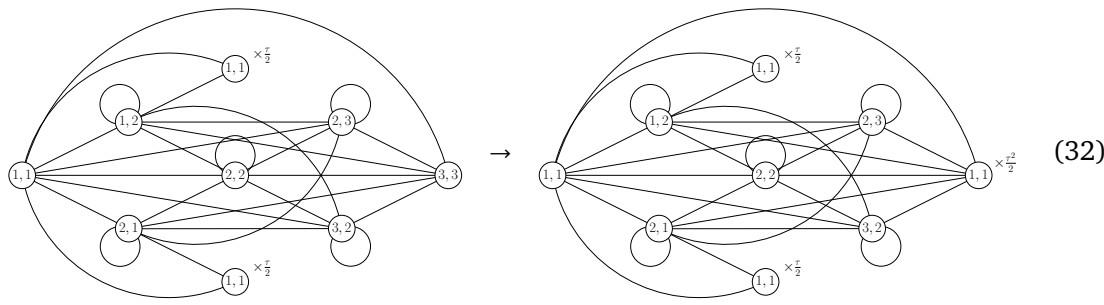

(32)

The above MPO now gives an approximation of $e^{\tau H}$ that captures all second-order terms exactly. Moreover, just as before, due to its size extensivity, it contains all higher-order terms that consist of disconnected first- and second-order parts.

This construction can be generalized to any order by the same idea, and the algorithm can be found in Alg. 1.

---

**Algorithm 1** Pseudocode for constructing the $N$'th order time evolution MPO

---

1: Inputs $\hat{H}, N, \tau$
2: $O \leftarrow \hat{H}^N$          ▷ multiply the hamiltonian $N$ times with itself
3: **for** $a \in [1, N]$ **do**
4:     $P \leftarrow$ permutations of $(1, 1, \ldots, 1, 3, 3, \ldots, 3)$ (3 occurs $a$ times)
5:     **for** $b \in P$ **do**
6:        $O[:, 1] = O[:, 1] + \tau^a \frac{(N-a)!}{N!} O[:, b]$
7:        Remove row and column $b$

---

In this section, we have explained our construction in terms of a single MPO tensor, but the construction is easily extended for systems with a non-trivial unit cell. For finite systems, one should impose the correct left and right boundary conditions:

$$L = \boxed{\begin{array}{c|c|c|c} 1 & 0 & \ldots & 0 \end{array}}, \qquad R = \boxed{\begin{array}{c} 1 \\ \hline 0 \\ \hline \ldots \\ \hline 0 \end{array}}.$$

(33)

# 4 Exact compression steps

The operator we arrived at in the previous section is essentially an operator-valued block matrix, a matrix where the entries correspond to operators. It is possible to multiply these by scalar-valued block matrices and in particular we can left and right multiply with the matrix

|       | (1,1) | (1,2)        | (2,1)        | (2,2) | (2,3) | (3,2) |
|-------|-------|--------------|--------------|-------|-------|-------|
| (1,1) | 1     |              |              |       |       |       |
| (1,2) |       | $1/\sqrt{2}$ | $1/\sqrt{2}$ |       |       |       |
| (2,1) |       | $1/\sqrt{2}$ | $-1/\sqrt{2}$ |      |       |       |
| (2,2) |       |              |              | 1     |       |       |
| (2,3) |       |              |              |       | 1     |       |
| (3,2) |       |              |              |       |       | 1     |

$$(34)$$

to obtain the MPO

|       | (1,1)                                          | (1,2) | (2,1) | (2,2)    | (2,3)    | (3,2)    |
|-------|------------------------------------------------|-------|-------|----------|----------|----------|
| (1,1) | $\mathbb{I} + \tau\,D + \frac{\tau^2}{2}\,DD$   | $C$   |       | $CC$     | $CD$     | $DC$     |
| (1,2) | $\tau\,B + \frac{\tau^2}{2}(DB+BD)$             | $A$   |       | $CA+AC$  | $CB+AD$  | $DA+BC$  |
| (2,1) | $\tau\,B + \frac{\tau^2}{2}(DB-BD)$             |       | $A$   | $CA-AC$  | $CB-AD$  | $DA-BC$  |
| (2,2) | $\frac{\tau^2}{2}BB$                            |       |       | $AA$     | $AB$     | $BA$     |
| (2,3) | $\frac{\tau^2}{2}B$                             |       |       |          | $A$      |          |
| (3,2) | $\frac{\tau^2}{2}B$                             |       |       |          |          | $A$      |

$$(35)$$

Given the boundary conditions at the left boundary [Eq. 33], there is no way to reach level (2,1). The corresponding entry in the left environments will always be zero and the corresponding row/column can therefore be safely removed.

Another way to see this compression is to look at the graphical representation of the original MPO, and noting the symmetry:

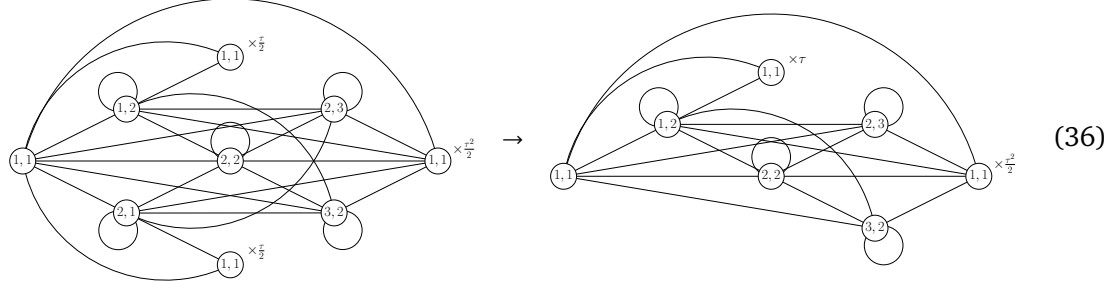

$$(36)$$

The transitions from (1,1) to (1,2) and from (1,1) to (2,1) are completely equivalent, we can therefore deform the diagram without changing the MPO. Simply add all arrows that leave the (2,1) node to the (1,2) node, and remove the (2,1) node.

A similar observation holds for the (2,3) and (3,2) nodes: all operators that follow the node before arriving at (1,1) are the same! We can redirect all arrows that point to (3,2), point them at (2,3) and remove the node (3,2). Equivalently, a similar basis transformation as in Eq. 34 will eliminate the transition from one of the (2,3) (3,2) levels back to (1,1). Given the right boundary condition [Eq. 33], the right environment will be zero for that level, and

the corresponding row/column can be removed.

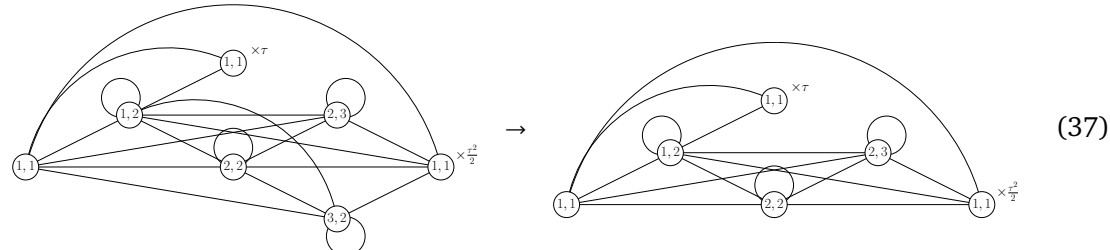

$$(37)$$

We eventually end up with the following operator:

|       | (1,1)                                            | (1,2) | (2,2)       | (2,3)                 |
|-------|--------------------------------------------------|-------|-------------|-----------------------|
| (1,1) | $\mathbb{I} + \tau\, D + \frac{\tau^2}{2}\, DD$  | $C$   | $CC$        | $CD + DC$             |
| (1,2) | $\tau\, B + \frac{\tau^2}{2}\, (DB + BD)$        | $A$   | $CA + AC$   | $CB + AD + DA + BC$   |
| (2,2) | $\frac{\tau^2}{2}\, BB$                          |       | $AA$        | $AB + BA$             |
| (2,3) | $\frac{\tau^2}{2} B$                             |       |             | $A$                   |

$$(38)$$

which represents a compressed version of the original second-order MPO in Eq. (31).

This exact compression step can be generalized to the general $n$'th order MPOs, see Alg. 2.

---

**Algorithm 2** Pseudocode incorporating exact compression

---
1: $O \leftarrow$ Alg. 1
2: **for** $c \in$ possible levels in $O$ **do**
3:      $s_c \leftarrow$ Sort the 1's in $c$ to the front
4:      $s_r \leftarrow$ Sort the 3's in $c$ to the front
5:      $n_1 \leftarrow$ the number of 1's in $c$
6:      $n_3 \leftarrow$ the number of 3's in $c$
7:      **if** $n_3 \leq n_1$ & $s_c \neq c$ **then**          ▷ Equivalent column
8:          $O[s_c, :] = O[s_c, :] + O[c, :]$      ▷ Add row $c$ to row $s_c$
9:          Remove row and column $c$
10:     **if** $n_3 > n_1$ & $s_r \neq c$ **then**          ▷ Equivalent row
11:         $O[:, s_r] = O[:, s_r] + O[:, c]$      ▷ Add column $c$ to column $s_r$
12:         Remove row and column $c$

---

# 5 Incorporating higher-order terms

At this point, we have found an MPO expression for $e^{\tau H}$ that is correct up to a given order $n$, but which also contains all disconnected higher-order terms that can be decomposed into smaller-order factors. Yet we can still incorporate more higher-order terms in the MPO without changing the bond dimension. Starting from the first-order MPO in Eq. 26, it was indeed noticed in Ref. [7] that the second-order term with Hamiltonians only overlapping on a single site can be readily included in the MPO. In this section, we show that our construction of the $N$th order MPO can be similarly extended to contain all terms of order $N + 1$, for which at least two out of the $N + 1$ composed Hamiltonian terms are such that one term ends on the same site as the other one starts. Terms that cannot be captured in this way are those that contain the composition of $N + 1$ contributions of the $A$ block on a given site.

The easiest way to understand this procedure is by studying the finite-state machines that generate $\hat{H}^N$ and $\hat{H}^{N+1}$, before turning it into the extensive zero-degree MPO that represents the exponential of $\hat{H}$ and applying any of the compression steps. Consider a certain path through the finite-state machine of $\hat{H}^N$, that starts at the $N$-tuple $(1,1,\ldots,1)$ and ends at the $N$-tuple $(3,3,\ldots,3)$. Within this path, we also want to systematically encode contributions coming from $\hat{H}^{N+1}$, namely contributions where at one particular site, which corresponds to one particular segment along the path, one term of the Hamiltonian stops and another term starts. This corresponds to having an extra mode 1 in the incoming tuple, and an extra mode 3 in the outgoing tuple. Note that this extra 1 and 3 can appear everywhere in the tuple, i.e. one operator term is forced to stop on this site and another term is started. The entry of the MPO representation of $\hat{H}^{N+1}$ corresponding to these extended tuples exactly contains the correct contribution to make this happen, so that we can add this contribution to the existing entry of $\hat{H}^N$ for the original values of these tuples. The other segments of the path through the finite state machine do not need to be changed. If for example the extra 1 and 3 appear on the same position in the extended tuples, this corresponds to an extra contribution of the on-site operator encoded in $D$, but that is certainly not the only possible contribution. We have to account for the fact that, after transforming this MPO into the extensive MPO using Algorithm 1, this contribution will be given the prefactor $\tau^N/N!$, whereas it should have a prefactor $\tau^{N+1}/(N+1)!$, which we can easily compensate by attributing it a proper factor. Furthermore, by adding an extra 1 and 3 in the incoming and outgoing tuples at all possible positions, identical configurations with multiple 1s and 3s in the extended tuples will be counted several times, namely exactly as many times as the number of 1s or 3s that appear in these extended tuples. This too is simply corrected for by dividing with these factors.

One final remark is that we only want to add these additional contributions to paths in the finite-state machine that already encode terms of $\hat{H}^N$ where the $N$ different factors overlap. This requires in particular that there are no 1s appearing in the right tuple, as this would indicate that some factors of $\hat{H}^N$ still have to start. Furthermore, if the left tuple would only contain 1s and one or more 3s, this corresponds to a contribution where some terms have already ended, and would count as a lower order contribution when building the extensive MPO. Hence, aside from the $(1,1,\ldots,1)$ tuple, all left tuples should contain one or more values 2.

The resulting algorithm is represented by the pseudocode in Alg. 3. The application of this extension step to the case $N=2$, followed by the transformation to the extensive MPO (Algorithm 1) and the compression step described in the previous section (which remains valid), gives rise to the following MPO[6]

| | (1,1) | (1,2) | (2,2) | |
|---|---|---|---|---|
| (1,1) | $\mathbb{I} + \tau D + \frac{\tau^2}{2}DD + \frac{\tau^3}{6}DDD$ | $C$ | $CC + \tau\{CCD\}$ | |
| (1,2) | $\tau(B + \tau\{DB\} + \frac{\tau^2}{2}\{BDD\})$ | $A$ | $2\{AC\} + \tau(\{ACD\} + \{BCC\})$ | $\cdots$ |
| (2,2) | $\frac{\tau^2}{2}(BB + \tau\{BBD\})$ | | $AA + \tau\{AAD\} + 2\tau\{ABC\}$ | |
| (2,3) | $\frac{\tau^2}{2}B + \frac{\tau^3}{6}(2\{BD\} + BD)$ | | $\frac{\tau}{3}(2\{AC\} + AC)$ | |

| | | (2,3) | |
|---|---|---|---|
| | (1,1) | $2\{CD\} + \tau\{CDD\}$ | |
| $\cdots$ | (1,2) | $2(\{BC\} + \{AD\}) + \tau(2\{BCD\} + \{ADD\})$ | (39) |
| | (2,2) | $2\{AB\} + \tau\{ABD\} + \frac{\tau}{3}\{BBC\}$ | |
| | (2,3) | $A + \frac{\tau}{3}(2\{AD\} + AD + 2\{BC\} + BC)$ | |

---

[6]Here we have introduced a shorthand notation for denoting the sum of all permutations $\sigma$ of a given set of $N$ operators as $\{X_1 X_2 \ldots X_N\} = \frac{1}{N!}\sum_\sigma X_{\sigma(1)} X_{\sigma(2)} \ldots X_{\sigma(N)}$. For example $\{BD\} = (BD + DB)/2$, $\{ABB\} = (ABB + BAB + BBA)/3$ or $\{ABC\} = (ABC + ACB + BAC + BCA + CAB + CBA)/6$.

---

**Algorithm 3** Pseudocode for the extension step

---

1: Inputs $\hat{H}, N, dt$
2: $O \leftarrow \hat{H}^N$
3: **for** $a, b \in$ possible levels in $O$ & $1 \notin b$ **do**   ▷ Incorporate higher order corrections
4:     if $2 \notin a$ & $3 \in a$, skip
5:     **for** $c, d \in [1 : N + 1]$ **do**
6:         $a_e =$ insert a 1 at position $c$ in $a$
7:         $b_e =$ insert a 3 at position $d$ in $b$
8:         $n_1 =$ the number of 1's in $a_e$
9:         $n_3 =$ the number of 3's in $b_e$
10:         $O[a, b] = O[a, b] + H^{N+1}[a_e, b_e] \tau \frac{N!}{(N+1)! n_1 n_3}$
11: Apply algorithm 2

---

# 6  Approximate compressions

There is one more possible compression for an $N$th order MPO, similar in spirit to the previous extension step. This compression step is only accurate up to order $N$, and it therefore slightly lowers the precision of the extended MPO. We will again illustrate the method starting from the second-order MPO, and then extend it to arbitrary order.

The essential observation is that the levels (12) and (23) in the second order MPO are similar. The diagonal elements $O_2[(12), (12)]$ and $O_2[(23), (23)]$ are equal in the lowest order in $\tau$. Furthermore, the transition from (12) to (11) and from (23) to (11) are also related. The lowest order of $O_2[(23), (11)]$ equals the lowest order of $O_2[(12), (11)]$, multiplied with an extra factor of $\frac{\tau}{2}$.

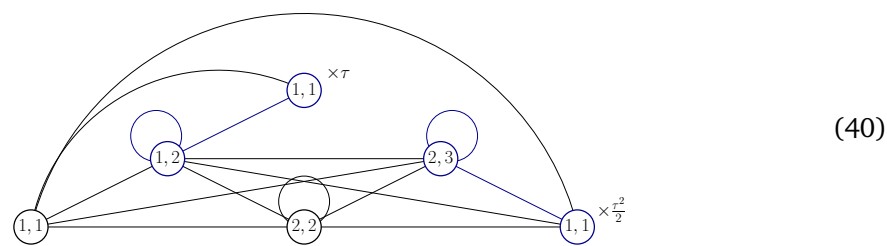

$$(40)$$

This means that one can add $\frac{\tau}{2}$ times the (23) column to the (21) column and remove the (23) level, and the resulting MPO will also be accurate up to second order! The second-order MPO now becomes

|  | (1,1) | (1,2) |
|---|---|---|
| (1,1) | $\mathbb{I} + \tau D + \frac{\tau^2}{2!} D^2 + \frac{\tau^3}{3!} D^3$ | $C + \tau\{CD\} + \frac{\tau^2}{2!}\{CDD\}$ |
| (1,2) | $\tau(B + \tau\{BD\} + \frac{\tau^2}{2!}\{BDD\})$ | $A + \tau\{AD\} + \frac{\tau^2}{2!}\{ADD\} + \tau(\{CB\} + \tau\{CBD\})$ |
| (2,1) | $\frac{\tau^2}{2!}(BB + \tau\{BBD\})$ | $\tau(\{AB\} + \tau\{ABD\}) + \frac{\tau^2}{2!}\{CBB\}$ |

...

...

|  | (2,2) |
|---|---|
| (1,1) | $CC + \tau\{CCD\}$ |
| (1,2) | $2(\{AC\} + \tau\{ACD\}) + \tau\{CCB\}$ |
| (2,1) | $AA + \tau\{AAD\} + 2\tau\{ACB\}$ |

$$(41)$$

Once again we can generalize this step to any order, as described in Alg. 4.

---

**Algorithm 4** Pseudocode for the approximate compression step

---

1: Apply algorithm 3
2: **for** $a \in$ possible levels in $O$ & $1 \notin a$ **do**
3:     $n_1$ = the number of 3's in $a$
4:     $b$ = replace all 3's with 1's in $a$
5:     $O[:, b] = O[:, b] + O[:, a] \tau^{n_1} \frac{(N - n_1)!}{N!}$
6:     remove level $a$

---

## 7  Numerical compression

In the previous three sections, we have provided analytical techniques for compressing and extending our construction for approximating $e^{\tau H}$ as an MPO. However, we can also compress the MPO numerically using singular-value decompositions. The idea behind this is that we interpret the MPO as a regular MPS with two physical legs, and truncate with respect to the 2-norm for states. This procedure should be taken with care, because we are working with a norm that is not suitable for operators, and should maybe only be used in cases where we can do *exact* compressions (for which the singular values are exactly zero, and it doesn't matter which norm is taken).

We can use this numerical compression for checking whether we have found all exact compression steps. If we do this on the uncompressed MPOs from Sec. 3, we observe that we indeed find a number of exact zero singular values in the MPO, corresponding to the analytical compression steps that we have identified above. After having done these analytical compressions, however, we find that the MPO cannot be compressed further. This shows that we have found all possible exact compressions.

## 8  Benchmarks

### 8.1  Precision of $n$th order MPO

Let us first illustrate the precision of our MPO construction. Therefore, we first optimize an MPS ground-state approximation $|\Psi_0\rangle$ of a given Hamiltonian $H$ in the thermodynamic limit and subsequently evaluate

$$p = |\lambda - iE_0 \delta t|, \qquad \lambda = \lim_{N \to \infty} \frac{1}{N} \log \langle \Psi_0 | U(\delta t) | \Psi_0 \rangle, \tag{42}$$

where $\lambda$ can be evaluated directly in the thermodynamic limit (see Sec. 2) and $E_0$ is the ground-state energy per site. In this set-up, we make sure that the MPS $|\Psi_0\rangle$ is approximating the true ground state quasi-exactly – in practice, we just take very large bond dimension – such that $p$ is indeed measuring errors in the MPO approximation $U(\delta t)$ for the time-evolution operator.

In Fig. 1 we plot this quantity as a function of $\delta t$ for both the $n$th-order MPO without extensions and approximate compressions and the extended and compressed MPO, each time for different orders. We find that the error has the expected scaling as a function of $\delta t$, showing that our MPO construction is correct up to a given order. We observe that the approximate compression and extension steps give rise to more precise MPOs, although the bond dimension is smaller. This shows that it is always beneficial to work with these MPOs.

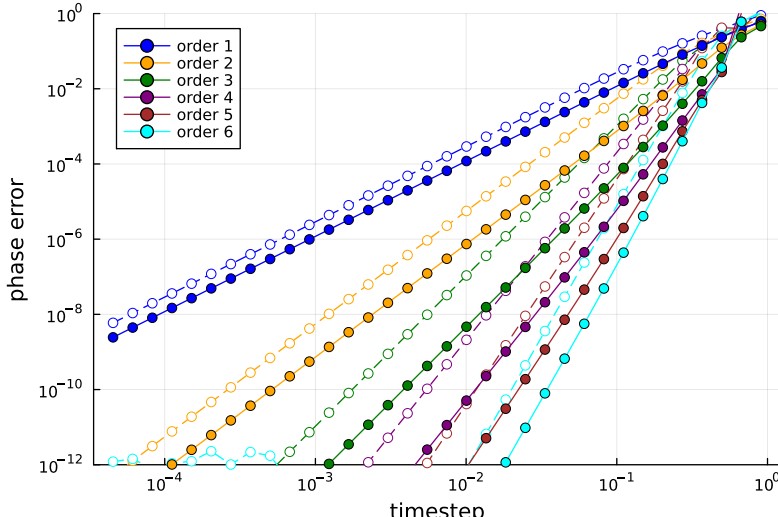

Figure 1: Precision of the $n$th order operator (as measured by Eq. 42) for the SU(2) symmetric spin-1 Heisenberg model, plotted as a function of the time step. We find the expected power-law behavior, where different orders directly correspond to different powers (the $n$th order operator has an error scaling as $n+1$). The open circles show the results from only using exact compression steps, while the filled circles were obtained using both the approximate compression and extension. Note that the fluctuations around $10^{-12}$ are due to limited numerical accuracy.

## 8.2 Efficiency

After having showed that our construction works as intended, we now show that it is actually efficient to use higher-order MPOs in practical MPS time-evolution algorithms. Let us therefore take the Hamiltonian of the two-dimensional transverse-field Ising model on a finite cylinder with spiral boundary conditions [Eq. (19)], find an MPS ground-state, perform a spin flip in the middle of the cylinder and time-evolve the state. This is the typical set-up for evaluating a spectral function. We time-evolve for a total time $T = 1$ with different times steps $\delta t$, where we approximate the time-evolution operator $e^{iH\delta t}$ by MPOs of different orders. In each time step, we perform a variational sweeping optimization of the new time-evolved MPS and keep the bond dimension fixed. After the time evolution we evaluate the fidelity per site with respect to a benchmark time-evolved state (which was obtained by the algorithm based on the time-dependent variational principle (TDVP) [23] with time step $\delta t = 0.0001$):

$$f = \frac{1}{N} \log \langle \Psi_{\text{bench}}(T) | \Psi_n(T) \rangle \, , \tag{43}$$

with $N$ the number of sites.

In the first panel of Fig. 2 we plot this fidelity density as a function of time step, showing that we indeed find higher precision with higher-order MPOs and that the error scales with the correct power of $\delta t$. Note that the first-order MPO is exactly the same as the $W_{II}$ operator from Ref. [7]. Curiously, we find that the error for the second-order MPO scales according to a third-order MPO, but this is not generically true and depends on the particular Hamiltonian.

In the second panel, we show the computational time as a function of the fidelity density, showing how much time is needed to reach a certain accuracy. This plot clearly shows that it is beneficial to go beyond the first-order MPO. For general models, we expect that we can obtain better fidelity at the same computational cost by using higher order methods. The extraordinary performance of the second-order MPO in this particular example originates from the fact that it is correct up to third-order, which is not expected in general.

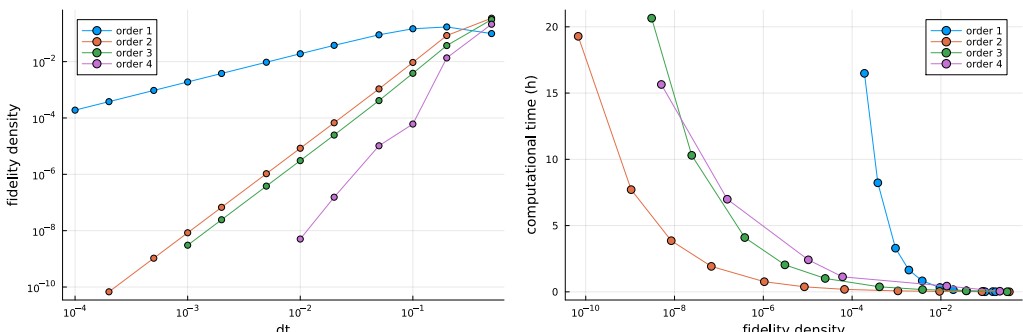

Figure 2: Benchmark results for the transverse-field Ising model on a cylinder ($W = 4$). In the left panel we plot the fidelity density [Eq. 43] as a function of time step. In the right panel we plot the fidelity density as a function of total simulation time.

## 8.3   Splitting schemes

There is a well known approach for generating higher-order time-evolution methods out of lower-order approximation schemes, by combining ingeniously chosen time steps [24]. Given a first order method, such as our time-evolution operator $O_1(t)$, it can be combined with alternating timesteps $t_1 = (1 + i)/2$ and $t_2 = (1 - i)/2$. The composite operator $O_1(t_1)O_1(t_2)$ $= O_2(t_1 + t_2)$ is then accurate up to second order [7]. In general, a second-order method and more than two time steps are required, in order to construct higher order schemes by combining only real time steps. This is also the basis behind higher order Suzuki-Trotter decompositions.

In contrast to these splitting schemes, the construction of the $N$th order MPO $O_N$ has a bond dimension as listed in the following table (where $\chi$ is the bond dimension of the $A$ block in the Hamiltonian):

| Order | Bond dimension |
|---|---|
| 1 | $1 + \chi$ |
| 2 | $1 + \chi + \chi^2$ |
| 3 | $1 + 3\chi + \chi^2 + \chi^3$ |
| 4 | $1 + 5\chi + 4\chi^2 + \chi^3 + \chi^4$ |

Even in the case that we assume that our MPO operators are fully dense, the composition of $N$ first-order operators will therefore always have a larger bond dimension than the construction we put forward.

Furthermore, for splitting schemes including complex-valued time steps, the resulting operator will exponentially grow high energy contributions before exponentially suppressing them in a subsequent step, raising serious concerns about their stability. These splitting schemes may however be useful as a trade-off between CPU time and memory usage. A high-order time evolution operator corresponds to an MPO tensor with an exponentially large bond dimension. By combining a splitting scheme with the highest-order operator that can still reside in memory, one can push time evolution simulations to even higher levels of accuracy.

## 8.4   Finite temperature

Our method can also be used to directly construct the finite temperature density matrix $e^{-\beta H}$, at different orders of precision. We have calculated the free energy and energy density for different values of $\beta$ at different expansion orders for the spin-$\frac{1}{2}$ XXZ model, directly in the thermodynamic limit (see Fig. 3).

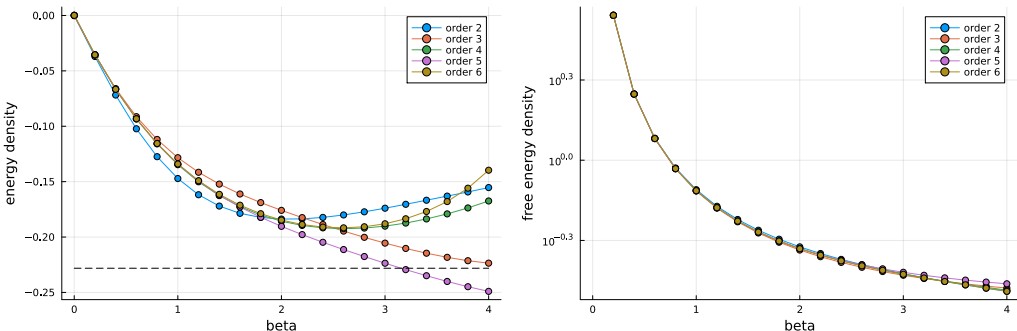

Figure 3: Finite temperature results for the spin-$\frac{1}{2}$ XXZ model in the thermodynamic limit. In the left panel we plot the energy density. The ground state energy density is indicated by a black dashed line. In the right panel we plot the free energy density.

This calculation is highly and straightforwardly parallelizable (at least on a shared-memory architecture), as it boils down to solving an iterative dominant eigenvalue problem of a block-sparse matrix. It is however fundamentally limited in the achievable $\beta$. At some crossover point (around $\beta \sim 2$ in this case) the error term will always start to dominate, and the results become wildly inaccurate. The best results will presumably be obtained by multiplying multiple density matrices at smaller $\beta$ (which can be calculated up to arbitrary precision).

# 9 Conclusion and outlook

We have introduced a new way of approximating the time evolution operator as an MPO correctly up to arbitrary order in the time step. The algorithm is formulated in the language of Hamiltonians represented as first-degree MPOs and is directly compatible with spatial symmetries (in particular, translation invariance) and non-abelian on-site symmetries. While such a construction is interesting in its own right, we have demonstrated that a higher order scheme allows us to speed up MPO time-evolution simulations by an order of magnitude – for a detailed comparison between MPO schemes and TDVP algorithms, we refer the reader to Ref. [17]. The higher-order MPOs can be readily used in existing time-evolution algorithms, leading to immediate speedups. For the reader's convenience, we have summarized the most useful MPO expressions in the Appendix.

It would be interesting to explore the interplay between the approximate compression step from Sec. 6 and the extension step from Sec. 5. The compression step should in principle introduce errors of order $N + 1$, while these are precisely the kind of terms we correctly try to incorporate in the extension step, and so in principle we would expect these steps to be at odds with each other. Nevertheless we observe that a combination of the two gives the best results, which is not yet fully understood.

In principle it is clear how one can apply a very similar methodology to time-dependent Hamiltonians. For the example of periodic driving, it could allow us to construct the time evolution operator over an entire period at once. In turn, we would be able to analyze this operator with spectral methods, extracting information on the effective time-averaged operator, as an alternative to the more conventional perturbative expansion.

In another direction, we expect that our MPO construction can be useful for optimizing other approximation schemes for the time-evolution operator. Notably, the use of efficient MPO representations can greatly benefit the classical optimization of quantum circuits for implementing dynamics on digital quantum simulators [25, 26].

## Acknowledgments

We would like to thank Bram Vanhecke and Frank Verstraete for earlier collaborations that inspired this work.

**Code availability**    The computer code can be found in the software package MPSKit.jl [27].

**Funding information**    MV and JH have received support from the European Research Council (ERC) under the European Union's Horizon 2020 program [Grant Agreement No. 715861 (ERQUAF)] and from the Research Foundation Flanders. LV is supported by the Research Foundation Flanders (FWO) via grant FWO20/PDS/115. IPM is supported by the National Science and Technology Council (NSTC) Grant Nos. 112-2811-M-007-044 and 113-2112-M-007-MY2.

## A    Explicit expressions

Here we recapitulate the expressions for the optimal first- and second-order MPOs. Starting from a Hamiltonian in MPO form

| $\mathbb{I}$ | $C$ | $D$ |
|---|---|---|
| | $A$ | $B$ |
| | | $\mathbb{I}$ |

the optimal first-order MPO is given by

| $\mathbb{I} + \tau D + \frac{\tau^2}{2}D^2$ | $C + \tau\{CD\}$ |
|---|---|
| $\tau(B + \tau\{BD\})$ | $A + \tau\{AD\} + \tau\{CB\}$ |

and the optimal second-order MPO is given by

| $\mathbb{I} + \tau D + \frac{\tau^2}{2}D^2 + \frac{\tau^3}{6}D^3$ | $C + \tau\{CD\} + \frac{\tau^2}{2}\{CDD\}$ | $CC + \tau\{CCD\}$ |
|---|---|---|
| $\tau(B + \tau\{BD\}$ $+ \frac{\tau^2}{2}\{BDD\})$ | $A + \tau\{AD\} + \frac{\tau^2}{2}\{ADD\}$ $+ \tau(\{CB\} + \tau\{CBD\})$ | $2(\{AC\} + \tau\{ACD\}) + \tau\{CCB\}$ |
| $\frac{\tau^2}{2}(BB + \tau\{BBD\})$ | $\tau(\{AB\} + \tau\{ABD\}) + \frac{\tau^2}{2}\{BBC\}$ | $AA + \tau\{AAD\} + 2\tau\{ACB\}$ |

The expressions for the higher-order MPOs are too large to display on this page, and we advise to implement the generic algorithms from the main text.

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
