# Peer review of "Efficient higher-order matrix product operators for time evolution"

_SciPost Physics, doi:SciPost Phys. 17, 135 (2024)_

## Round 1 · Referee Report · Anonymous (Referee 1) · 2023-5-12

Strengths

-provides a novel efficient algorithm for representing powers of operators as compact MPOs

Report

In spite of significant progress over the past few years, the study of the time evolution of quantum systems using matrix product states still holds many formidable challenges. An important element in this context is the search of efficient representations of time-evolution Hamiltonians in terms of matrix product operators, a task which could in principle become extremely costly if one wishes to go beyond the lowest-order algorithms.

Through some careful inspection of the block structure of the resulting MPOs, the authors manage to obtain a recipe for systematically codifying higher orders of powers of generic model Hamiltonians in an efficient way, allowing for higher-order calculations of the time evolution operators ~exp(tH).
The key of this algorithm is a clever grouping of terms, collecting iteratively non-overlapping (or, when going to higher orders, with limited overlap) operators to each order, which can then be encoded efficiently.

Interestingly enough, the authors find that for a concrete example they consider (the transverse-field Ising model on a cylinder), the second-order expansion turns out to be equivalent to the third-order one. While this is somewhat of a nice surprise, it seems to be in some way at odds with the authors’s statement that evidence from “numerical” compression, obtained by performing singular-value decompositions of the MPO tensors, suggests that this encoding is the most efficient possible: in this case, it seems that the third-order prescription would not really be the most efficient one.
I understand that this feature, as the authors clearly state, seems to be model-specific, I simply wonder if this property is shared by a family of models/geometries, and if so, whether for this kind of family an even more efficient representation could be derived.

It is also not entirely clear to me from the presentation in Fig.1 what is the individual effect of the “approximate” compression presented in Sec.6: the authors claim that the result is more accurate with less bond dimension required, but from what I understand, the results presented combine also another resummation together with it, so it’s hard for me to tell what is the individual effect of this approximation - perhaps the authors could comment a bit more on that.

Another minor point is that I don’t understand the meaning of the open light blue circles in the bottom-left corner of Fig.1, which seem to deviate completely from the other trends.

Aside from these remarks, I think the paper is well-written and of interest to the community, so I recommend its publication.

Requested changes

-add a few comments on the issues discussed in the report

  • validity: high
  • significance: good
  • originality: good
  • clarity: good
  • formatting: excellent
  • grammar: excellent

Author:  Laurens Vanderstraeten  on 2024-07-03  [id 4599]

(in reply to Report 1 on 2023-05-12)

We would like to thank the Referee for their positive report on our work. Below we respond to the two comments that were raised by the Referee:

1- Indeed, this was a surprising and unexpected result! We have attempted to gain some intuition for when this property holds, because it opens the door to even more efficient simulations for a potentially larger set of models! It is sometimes possible to show that the error must be an even power of the time step using symmetry considerations, but we did not find anything obvious in this case. The SVD analysis was done with a random hamiltonian to prevent exactly these kind of "special occurances".

2- The approximate compression applied to an MPO that is accurate up to dt^N introduces an error of order dt^(N+1), but one which barely shows up in the log-log plot. The approximate expansion step keeps the bond dimension the same, and introduces some corrections of order dt^(N+1). We have attached a plot where we redid the analysis in three different cases, for N = 1,2,3 and 4. The solid line is the "plain" option, where we did neither the approximate extension nor the approximate compression. The dashed line was obtained by only doing the approximate extension and the dash-dotted line was obtained by doing both the approximate extension and approximate compression. It is difficult to distinguish between the last two curves, as they overlap almost entirely. There are some noticeable but small differences at higher N.

3- These are below the numerical noise floor, and should in principle be disregarded. We have added a note in the caption of this figure.

Attachment:

in_depth.pdf

---

## Round 1 · Referee Report · Anonymous (Referee 2) · 2023-6-17

Report

The manuscript presents new ideas on how to approximate the time evolution operator of a quantum many-body Hamiltonian using tensor networks. Building on previous works, the authors identify a systematic way of constructing compact MPO representations of the mentioned operator that are exact up to a given order. Furthermore, by cleverly grouping the terms included in the expansion they construct an algorithm that is suitable to work on the thermodynamic limit.

While the subject itself is technical, the paper is well-writen, contains practical benchmarks and is of interest to the computational physics community, so I recommend its publication.

Requested changes

1- It would be interesting to compare the computational times needed to obtain a given fidelity in Fig. 2 with the scheme presented in this paper to present methods in the TN literature. In the introduction, the authors argue that the methods that rely on integrating Schrödinger's equation using small time steps require investing many CPU hours. The authors could compare the times needed with the scheme presented here to the times that would be needed with TDVP, as they use that method to obtain the benchmark state.

2- Similarly, the authors could add a line with the (analytically or numerically) exact results in Fig. 3 to increase the clarity of the presentation.

  • validity: high
  • significance: high
  • originality: good
  • clarity: good
  • formatting: excellent
  • grammar: excellent

Author:  Laurens Vanderstraeten  on 2024-07-03  [id 4598]

(in reply to Report 2 on 2023-06-17)

We would like to thank the Referee for their positive report on our work. Below we respond to the two comments that were raised by the Referee:

1- We would like to avoid making direct comparisons with TDVP for the computational time, because this complicates the analysis. There are different parameters that one can fiddle with (what is the precision of the integrator, does one keep AL/AR consistent with each other during the time evolution, how precisely does one canonicalize the states, what does one parallelize over, etc.), and they will all result in slightly different performance. The rationale of comparing computational time for different MPO time evolution simulations is that they all make use of the same code, and whatever improvements one makes to the code should only result in a global improvement. Moreover, the comparison between TDVP and MPO-evolution has been done in great detail in Ref. 17, which shows that it strongly depends on the application which method is the most efficient. We have added a reference to this review paper in the conclusion to make this point explicit.

2- We agree that we could provide exact results for these quantities in Fig. 3, but we believe that this distracts from the message we want to convey with this figure: We want to primarily convey the point that our series expansion unavoidably breaks down at some value of beta, in the sense that increasing the order does not help.

---

## Round 2 · Referee Report · Anonymous (Referee 1) · 2024-7-18

Report

The authors have (slightly) improved the manuscript, which was already of good quality. I recommend its publication.

Recommendation

Publish (meets expectations and criteria for this Journal)

---

## Round 2 · List of Changes

1- We have added an explicit reference to a review paper on time evolution schemes for MPS, in order to make the comparison of our MPO scheme with other schemes more obvious. 2- We have expanded the caption of Fig. 1. 3- We have changed Sec. 5 to make the discussion clearer.

---

## Editorial Decision

published